# The Impact of Social Determinants of Health on Australian Women’s Capacity to Access and Understand Health Information: A Secondary Analysis of the 2022 National Women’s Health Survey

**DOI:** 10.3390/healthcare12020207

**Published:** 2024-01-15

**Authors:** Neve Davison, Karin Stanzel, Karin Hammarberg

**Affiliations:** Global and Women’s Health, School of Public Health and Preventive Medicine, Monash University, Melbourne, VIC 3004, Australia; neve.davison@monash.edu (N.D.); karin.hammarberg@monash.edu (K.H.)

**Keywords:** health literacy, social determinants of health, women’s health, healthcare, health information seeking

## Abstract

The ability to access, understand, judge, and use health information is crucial for making informed decisions about health and optimal health outcomes. This secondary data analysis investigated associations between social determinants of health and Australian women’s ability to access and understand health information using data from 10,652 women who responded to the 2022 National Women’s Health Survey. A score (0–5) was created based on five questions assessing the participants’ ability to access and understand health information, which was dichotomised into low (≤3) and high (≥4) scores. The data were analyzed using descriptive statistics, univariate comparisons, and multivariable binary logistic regression. Almost a quarter of the women had a low score. Non-native English speakers were approximately four times more likely to have low health literacy than native English speakers. Additionally, women without tertiary education, financially disadvantaged women, and First Nations women were almost twice as likely to have lower health literacy than other women. These findings suggest that social determinants of health decrease the capacity to access and understand health information. To reduce health inequalities, healthcare systems and health professionals must consider the factors that reduce women’s capacity to access and understand health information and address the health information needs of socioeconomically disadvantaged women.

## 1. Introduction

The social determinants of health are described by the World Health Organization as the “conditions in which people are born, grow, live and age” that have an impact on health outcomes [1] (p. 357). Social determinants associated with poor health outcomes include belonging to a cultural or linguistic minority group, having limited education, and being financially disadvantaged [2]. 

Social determinants of health have been linked to the attainment of health literacy skills. An Australian cross-sectional study including 813 participants investigated the association between the social determinants of health and health literacy [3]. The authors reported a negative association between socioeconomic status and health literacy [3].

In 2012, Sorensen and colleagues [4] defined health literacy as an “individual’s ability to access, understand, appraise, and apply health information in order to make judgments and make decisions in everyday life concerning health, health behaviour and healthcare” [4]. In 2014, the Australian Commission on Safety and Quality in Health Care expanded the definition and described two components of health literacy: ‘individual health literacy’ and ‘the health literacy environment’ [5]. Individual health literacy refers to the ability to access, understand, appraise, and apply health information in order to improve or maintain health. The health literacy environment refers to the health system, which includes policies, infrastructure, processes, materials, and people, and its impact on individuals’ capacity to access, understand, appraise, and use health-related information and services [5]. 

A systematic review of 96 studies investigating health literacy and health outcomes conducted by Berkman and colleagues (2011) [6] concluded that people with poor health literacy are less likely to use preventive healthcare services and screening measures, such as mammograms, cervical screening, and vaccinations. Hence, a lack of preventive care may explain why people with low health literacy are at increased risk of hospitalizations and emergency care admission [6,7]. Hospitalizations for potentially preventable conditions increase the burden on the healthcare system and the cost of healthcare [8]. 

In 2006, the Australian Bureau of Statistics (ABS) released the results of the Adult Literacy and Life Skills Survey (ALLS). The ALLS provided information about whether Australians’ literacy skills were adequate for the challenges they face in work and daily life. The ALLS data included a section measuring health literacy. Health literacy was defined as “the knowledge and skills required to understand and use information relating to health issues such as; drugs and alcohol, disease prevention and treatment, safety and accident prevention, first aid, emergencies, and staying healthy”. The ALLS assessed functional aspects of literacy, such as understanding text, finding information in documents, and problem-solving capabilities. The report revealed that approximately three in five Australian women have low health literacy [9]. Furthermore, it was revealed that the proportion of people with low health literacy is even higher in the most disadvantaged groups. For example, at that time, 84% of unemployed people and 74% of migrants born in non-English speaking countries had low health literacy [9]. 

In 2018, the ABS conducted the Health Literacy Survey (HLS) [10]. The HLS moves beyond the functional approach and covers nine conceptionally distinct areas of health literacy, including feeling understood and supported by healthcare providers; having sufficient information to manage one’s health; actively managing one’s health; social support for health; appraisal of health information; ability to actively engage with healthcare providers; navigating the healthcare system; ability to find good health information; and understanding health information well enough to know what to do [10]. The survey found that one in four respondents were able to find good health information easily, 11% reported that they always found it easy to appraise health information, and less than one in four strongly agreed to having sufficient information to manage their health [10]. Although the information about health literacy provided in the HLS is not comparable to the ALLS data, it showed that there are gaps in health literacy skills in the Australian population which are likely to have adverse effects on health outcomes. 

The existing literature investigating the influence of some social determinants of health on health literacy is limited. The level of health literacy among Aboriginal and Torres Strait Islander peoples, Australia’s First Nations population, which represents 3.2% of the Australian population, has yet to be reported; however, it is assumed to be worse than the health literacy of non-Indigenous peoples [11,12]. Additionally, health literacy has been shown to differ between men and women [4]. Studies show that women generally have better health literacy or the same level of health literacy as men [9,10,13]. Little is known about how the social determinants of health influence the health literacy of women in Australia. Therefore, the aim of this study was to investigate the influence of social determinants of health on Australian women’s capacity to access and understand health information. 

## 2. Materials and Methods

### 2.1. Participants and Recruitment

Jean Hailes for Women’s Health is an Australian government-funded not-for-profit organization that aims to improve women’s health by providing free, easy-to-understand health information for all women, girls, and gender-diverse people [14]. Annually, the Jean Hailes National Women’s Health Survey is used to gauge Australian women’s health, health behaviors, and health information needs [14]. The findings are used to inform health-promoting activities that aim to meet the needs of Australian women.

The 2022 Jean Hailes Nation Women’s Health Survey [14] collected data from March to May through an anonymous online survey. A total of 14,407 responses to the survey were recorded. Of these, 11,006 respondents completed the survey, provided logical responses, were over the age of 18, and did not identify as male [14]. To reach a diverse group of women, the survey was translated into Chinese (simplified), Arabic, and Vietnamese [14] by a translation service. For quality assurance of translation accuracy, this service follows the Australian Government Language Services Guidelines [15]. These guidelines recommend the use of independent checking, which refers to the procedure where one National Accreditation Authority for Translators and Interpreters (NAATI)-credentialled translator translates the document and a second NAATI-credentialled translator independently revises the translated product against the English original. Respondents were recruited through multiple communication channels, including social media, the Jean Hailes website, and email lists. Additional recruitment was completed in partnership with national community organizations and organizations that represent women from culturally and linguistically diverse groups [14]. There was targeted recruitment of Aboriginal and Torres Strait Islander women to gain a representative sample of this population [14].

### 2.2. 2022 Jean Hailes National Women’s Health Survey

The 2022 National Women’s Health Survey consisted of 30 questions related to Australian women’s health, health behaviors, and health information needs [14]. This secondary data analysis conducted for this study included questions relating to respondents’ demographic characteristics and capacity to access and understand health information. The demographic characteristics were age, level of education, self-reported health (SRH), financial situation, whether the individual identified as Aboriginal and/or Torres Strait Islander, main language spoken at home, country of birth, and time in Australia. SRH was gauged by asking: “How would you describe your health?” (excellent/very good/good/poor/very poor). The questions: “What language do you mainly speak at home” and “In which country were you born?” were asked to ascertain the participants’ main language and country of birth. The women who were born outside of Australia were asked the follow-up question: “How many years have you lived in Australia?” Their socioeconomic circumstances were gauged using two questions: “What is the highest level of education you have completed?” and “How would you describe your financial situation?” (living comfortably/doing alright/just getting by/finding it quite difficult/finding it very difficult). To ascertain the number of women identifying as Aboriginal or Torres Strait Islanders, the question “Are you of Aboriginal or Torres Strait Islander origin?” was asked.

The capacity to access and understand health information was assessed through two questions. The first question asked the respondents how many of the four following statements applied to them (Yes/No): “I know how to access the health services I need,” “I can easily find health information in my language,” “I understand most of the information my doctor or other health professionals tells me,” and “I feel confident asking my doctor or other health professionals questions when I don’t understand something.” The second question related to internet use and navigation skills, asking respondents “Are you able to use the internet to find information and make video calls easily?” (yes/no). 

### 2.3. Ethics Approval

The 2022 Jean Hailes National Women’s Health Survey was approved by the Bellberry Human Research Ethics Community [2018-03-187-A-10], March 2021. Separate ethics approval was not sought for this secondary data analysis study. All the data used were unidentifiable and stored securely on the Monash University Network.

### 2.4. Data Management

Women who responded ‘Prefer not to answer’ to questions assessing their capacity to access and understand health information or questions relating to the demographic characteristics of interest were excluded from the analysis.

#### 2.4.1. Health Literacy Score

A score was developed using the responses to the five statements that gauged the two components of health literacy: accessing and understanding health information. Each of the five items was scored 1 if the response was ‘Yes’ and 0 if the response was ‘No,’ giving a score with a possible range of 0–5, where 0 indicates the lowest and 5 the highest capacity to access and understand heath information. Although this study-specific score does not capture all components of health literacy, for brevity, it will be referred to as the health literacy score in this paper. The reliability of the score was checked to determine the internal consistency. Briggs and Cheek (1986) [16] recommend that the appropriate range for mean inter-item correlation is 0.2 to 0.4. The mean inter-item correlation for this health literacy score was 0.26.

#### 2.4.2. Dichotomized Variables

##### Health Literacy Score

To test for associations between social determinants of health and the capacity to access and understand health information, the health literacy score was dichotomized into low (≤3) and high (≥4) health literacy. This split was determined based on current health literacy literature and the distribution of the data [6].

##### Time in Australia

The two categories for time in Australia were based on a report released by the Department of Immigration and Citizenship, which classified ‘new arrivals’ as people who had arrived in Australia within the previous five years [17]. Therefore, the two categories for time in Australia used in this secondary data analysis were ≤5 years and >6 years since arriving in Australia.

##### County of Birth and Main Language

The country of birth was dichotomized into English-speaking or non-English-speaking countries. This classification was adapted from the ABS [18] and described Australia, the United Kingdom (UK), The Republic of Ireland, the USA, Aotearoa New Zealand, Canada, South Africa, British Overseas Territories, and the US Virgin Islands as the main English-speaking countries. The main language was dichotomized to align with the country of birth.

##### Other Dichotomized Variables

The education level was dichotomized as having tertiary education and no tertiary education. This was based on the Australian Qualification Framework [19]. SRH was dichotomized as excellent/very good and good/poor/very poor, following the method used by the ABS for dichotomizing this variable in the 2018 National Health Survey [10]. The participants’ financial situation was dichotomized as living comfortably/doing alright and just getting by/finding it quite difficult/finding it very difficult. First Nations origin was also dichotomized as Aboriginal, Torres Strait Islander, both, or not First Nations.

### 2.5. Statistical Analyses 

The data were analyzed using SPSS Statistics Version 28 [20]. Descriptive statistics were performed for the respondent demographics characteristics using frequencies and percentages or means and standard deviations where required. Following the descriptive statistics, a univariate analysis was performed. Contingency tables, chi-square tests, and independent-samples t-tests were used to analyze differences in the dichotomized health literacy score across the demographic characteristics. 

Significant results were included in a multivariable binary logistic regression model, which aimed to identify the factors that contributed to low health literacy. The country of birth and time in Australia were excluded from the regression analysis. The country of birth was excluded because it co-varied with the main language. The time in Australia was excluded because it only applied to those who were born in a country other than Australia.

## 3. Results

### 3.1. Respondents’ Characteristics

Of the 11,006 women in the data set, 354 (3.2%) were excluded from the analysis. Of these, 98 (27.7%) were excluded for responding “prefer not to answer” to one or more of the questions used to construct the health literacy score, and 256 (72.3%) were excluded for responding “prefer not to answer” to one or more of the questions related to the investigated social determinants of health. The questions about the participants’ financial situation (*n* = 90, 25.4%) and whether they were of First Nations origin (*n* = 76, 21.4%) were the most common questions not responded to.

The demographic characteristics of the remaining 10,652 women are displayed in Table 1. The sample was broadly representative of the Australian female population. However, the proportion of women with tertiary education was more than three times that of the general population, and the proportion of non-English speakers was about one-fifth of the proportion of non-English speakers in the general population. One in five respondents were born outside of Australia. Of these, just over half were born in countries where English is the main language. Additionally, the proportion of First Nations women was slightly higher than the proportion of First Nations women in the general population. This was likely due to Jean Hailes for Women’s Health implementation of specific recruitment strategies to target First Nations women.

### 3.2. Social Determinants That Influence the Capacity to Access and Understand Health Information

Just under a quarter of the respondents had a low capacity to access and understand health information (score ≤ 3). Comparisons across demographic characteristics showed significant differences between women with low and high health literacy scores (Table 2). A greater proportion of women with low health literacy scores came from more disadvantaged backgrounds.

Almost one-third of the women without a tertiary education had low health literacy scores. This was nearly twice the proportion of women with low health literacy scores who had completed their tertiary education. Over a third of the women who reported being financially disadvantaged had low scores. The proportion of First Nations women with low health literacy scores was almost double that of women who did not identify as Aboriginal or Torres Strait Islander.

All the measures of cultural and linguistic diversity showed that a greater proportion of women from these communities had low health literacy scores. Half of the women who were recent arrivals (lived in Australia for ≤5 years) and over half of the women whose main language was not English had low health literacy scores.

### 3.3. Social Determinants Associated with a Low Capacity to Access and Understand Health Information

The variables included in the multivariable binary logistic regression analysis were age, education, SRH, financial situation, First Nations origin, and main language. These variables were significant in the univariate analysis (*p* ≤ 0.05).

The results of the multivariable binary logistic regression model are shown in Table 3. The respondents whose main language was not English were almost four times more likely to have low health literacy scores than the women who spoke English as their main language. Additionally, women without a tertiary education, who were financially disadvantaged, or who identified as First Nations were approximately two times more likely to have low health literacy scores than women with a tertiary education who were financially well resourced or women who did not identify as First Nations women, respectively.

## 4. Discussion

This study is one of the first to assess the influence of social determinants of health on Australian women’s capacity to access and understand health information. The findings indicate that some of the social determinants of health limit women’s ability to access and understand health information. The findings suggest that speaking a language other than English at home, not having a tertiary education, being financially disadvantaged, and being of Aboriginal or Torres Strait Islander origin were all associated with a diminished ability to access and understand health information.

One of the strengths of this study was its size and wide reach. The 2022 National Women’s Health Survey had over 11,000 respondents, and 10,652 were included in this secondary data analysis. It was the first to be accessible to women from across Australia that assessed the influence of social determinants of health on health literacy capacity. Furthermore, to recruit women from marginalized populations, including First Nations women, targeted recruitment methods were used. As a result, First Nations women were over-represented, enabling a better understanding of the health literacy skills and potential barriers to accessing and understanding health information experienced by this minority population. 

However, we acknowledge some limitations. Only women with internet access and digital literacy skills were able to complete the online survey. This was a secondary data analysis, and the survey was not specifically designed to assess health literacy. Consequently, the analysis to investigate whether social determinants of health are linked to health literacy was restricted by the available survey questions, which did not include a validated instrument to measure health literacy. Therefore, care must be taken when interpreting these findings, as only the capacity to access and understand health information could be assessed. Additionally, although the five statements used to create the health literacy score assessed practical tasks, they were self-assessments. This means that the health literacy scores created from the five statements reflect self-perceived health literacy skills and abilities rather than an objective assessment of how Australian women access and understand health information.

Socioeconomic circumstances influence people’s ability to access and understand health information. This study used education and financial situation as proxies for socio-economic status. Education helps people attain fundamental literacy skills including reading, listening, speaking, and writing, which provide the stepping stones to the development of health literacy skills [22,23]. Previous large cross-sectional studies investigating the association between sociodemographic characteristics and health literacy reported that lower levels of education were associated with a greater likelihood of having inadequate health literacy [24,25,26]. Similar findings were obtained in this study, as the women without a tertiary education were almost two times more likely to have a reduced ability to access and understand health information than the women who had a tertiary education. 

Evidence indicates that being financially disadvantaged is associated with an increased likelihood of having inadequate health literacy. Supporting this was the finding in this study that financially disadvantaged women were more likely to have low health literacy scores than their financially advantaged counterparts. Both this study and Svendsen et al.’s 2020 study reported an approximately twofold increase in the likelihood of low health literacy among financially disadvantaged people [26,27].

Culture and language are determinants of health that have been shown to influence health literacy [3,4]. Factors such as a lack of information published in the preferred language, inaccessible or difficult-to-navigate websites, or information that may not be culturally appropriate or sensitive can act as barriers to health literacy [28,29]. This study found that the group whose main language was not English or who had recently arrived in Australia had the highest proportion of women with a low health literacy score. Women whose main language was not English were almost four times more likely to have low health literacy than those with English as their main language. Over half of the women whose main language was not English reported that they could not access information in their preferred language. Additionally, women who had recently arrived in Australia were twice as likely to have low health literacy scores than women who were born in Australia or had lived in Australia for more than six years. Similar findings have been reported globally [30,31]. Bergman et al. (2021) [30] investigated the health literacy and e-health literacy of Arabic-speaking migrants in Sweden. They found that Arabic-speaking migrants had significantly lower health literacy than Swedish speakers and that migrants who had been living longer in Sweden reported higher health literacy than those who had settled recently. A similar study conducted in Denmark by Fredsted Villadsen et al. (2020) explored the health literacy and e-health literacy of pregnant immigrant women, their descendants, and Danish-born women [31]. The authors found that migrant women had lower health literacy and e-health literacy than Danish-born women [31].

Globally, information on the health literacy of Indigenous people is limited. Although no studies have assessed the level of health literacy within the Aboriginal and Torres Strait Islander population, the health literacy of Indigenous Australians is thought to be lower than that of non-Indigenous Australians [10]. This study was the first Australian study to report on the level of health literacy of Aboriginal and Torres Strait Islander women. It used targeted recruitment methods to gain a representative sample of First Nations women to investigate their health literacy. Although this study only assessed access to and understanding of health information, it found that over 40% of First Nations women had low health literacy scores and were almost twice as likely to have low scores than non-Indigenous women. Aboriginal and Torres Strait Islander peoples are a particularly vulnerable group within the Australian population [32,33]. They have a lower life expectancy, a greater burden of disease, and less access to healthcare services compared to non-indigenous Australians [32,34]. Improving health literacy within this population might be a step towards improving health outcomes. 

Aotearoa New Zealand is the only country to report the health literacy of its Indigenous population [35]. In a national survey, the Adult Literacy and Life Skills Survey (ALLS) was used to assess health literacy in the Māori population. The ALLS was part of an international study coordinated by Statistics Canada and the Organisation for Economic Cooperation and Development that assessed adult literacy and numeracy skills across the four domains of prose literacy, document literacy, numeracy, and problem-solving [9]. Health literacy skills were assessed across these domains, and a cut-off score was used to identify the proportion of people with inadequate health literacy skills [9]. Approximately 75% of Māori women had inadequate health literacy skills, which was significantly higher than the 56% recorded in the general adult New Zealand population. Although comparisons between this study and the results of the ALLS cannot be made, both studies found that Indigenous women have lower health literacy scores than non-Indigenous women. This likely contributes to their poorer health outcomes. 

Understanding the barriers to and enablers of health literacy can inform strategies to improve women’s access to and understanding of health information. In addition to the sociodemographic factors that determine access to health information, the available health information resources can be inaccessible due to the density of their text and the use of difficult-to-understand language [29]. Hence, the health literacy environment, including health systems, health policies, and resources must better meet the needs of people with low health literacy. 

Inadequacy of the healthcare system component of health literacy in part explains the findings of this study. For most people, their first contact with the healthcare system is through their general practitioner (GP) [36]. However, approximately one in five women in this study reported not knowing how to access the health services they needed, which may include GPs and other primary healthcare providers. Similarly, studies have reported that people with unmet healthcare needs are more likely to have poor health literacy [37]. This suggests that easy-to-understand information on how to access health services is essential. How to navigate healthcare systems, such as making appointments with primary healthcare providers, should be promoted in communities of newly arrived migrants, in particular, for people from non-English speaking backgrounds and financially disadvantaged communities, as over a third of women from these communities responded ‘No’ when asked if they could access the health services they needed.

Social determinants of health can impact an individual’s ability to access and understand health information, including information provided by healthcare providers. Practitioners need to be aware that vulnerable groups may lack the skills needed to find and understand health information. To meet the needs of these groups, the health literacy environment must be adapted to be accessible for all levels of literacy and health literacy competency. Strategies need to go beyond text-based information and include a broad range of communication tools, including the use of different modes of health information resources such as podcasts, videos, and animations, and ensure that information is available and accessible in multiple community languages [38,39]. Additionally, to potentially mitigate health inequalities in minority populations, such as culturally and linguistically diverse groups and First Nations peoples, health information and health literacy interventions should be developed using community involvement and co-design [32,33].

This study used a study-specific measure of health literacy that assigned a score to reflect the participants’ capacity to access and understand health information. This measure did not assess all components of health literacy and relied on participant self-reporting. A number of validated instruments exist that measure health literacy more comprehensively. The Newest Vital Signs (NVS), the Test of Functional Health Literacy in Adults (TOFHLA), the Rapid Estimate of Adult Literacy in Medicine (REALM), and the HLQ are examples of instruments that have been used to assess health literacy [40,41,42]. The NVS is a short, easy-to-complete instrument that assesses a person’s understanding of a nutrition label. The TOFHLA measures an individual’s ability to read and comprehend the materials they may receive in a healthcare setting, and the REALM measures word recognition and the ability to decode and pronounce words [42]. These assessment tools focus on measuring an individual’s ability to understand health information and do not assess their ability to access, appraise, or apply this information. In contrast, the HLQ evaluates self-reported health literacy needs and abilities [40,41]. As a result, studies have proposed that health literacy assessment tools measure individuals’ perceptions and views about health literacy rather than actual knowledge and consequently are subjective [43]. This highlights the need for a validated instrument that objectively assesses both components of health literacy—an individual’s health literacy and the health literacy environment.

Future research should use a validated measure of health literacy to assess the influence of the social determinants of health on health literacy within the Australian population, including specifically in minority populations such as culturally and linguistically diverse communities and First Nations peoples. One recommendation is that future versions of the National Women’s Health Survey could use a validated measure of health literacy that assesses all components of health literacy. Additionally, neither the 2006 Australian ALLS [9] nor the 2018 Australian National Health Survey [10] reported on the health literacy of First Nations peoples, and no other study has measured the health literacy of First Nations peoples. Hence, research gauging the health literacy of First Nations peoples is needed to identify targets for interventions that may improve health literacy and avoid interventions that have previously been unsuccessful [44].

## 5. Conclusions

This study is the first to provide evidence of the effects of social determinants on the capacity of women from across Australia to access and understand health information. It shows that social determinants of health such, as limited English language proficiency, limited financial resources, not having a tertiary education, or identifying as an Aboriginal or Torres Strait Islander, reduce women’s capacity to access and understand health information.

Future research should investigate the influence of social determinants of health on the four components of health literacy: accessing, understanding, appraising, and applying health information. The health literacy of the Australian population, particularly Aboriginal or Torres Strait Islander peoples, should be measured to inform the development of interventions to improve health literacy. Ultimately, understanding the influence of social determinants of health on health literacy will help improve the health outcomes of all Australian women.

## Figures and Tables

**Table 1 healthcare-12-00207-t001:** Respondents’ characteristics compared to the Australian general population.

	Total (*n* = 10,652)	General Population *
	M (SD)	Median
Age (years)	50.55 (14.9)	39
	*n* (%)	%
Education		
No tertiary education	3978 (37.3)	78.4
Tertiary education	6674 (62.7)	21.6
Self-reported health ^1^		
Good/poor/very poor	6220 (58.4)	43.6
Excellent/very good	4432 (41.6)	56.4
Financial situation		
Just getting by/finding it quite difficult/finding it very difficult	2463 (23.1)	37
Living comfortably/doing alright	8189 (76.9)	63
First Nations origin		
First Nations	507 (4.8)	3.2
Not First Nations	10,145 (95.2)	96.8
Main language		
Language other than English	435 (4.1)	22
English	10,217 (95.9)	78
Country of birth		
Born overseas	2124 (19.9)	27.6
Australia	8528 (80.1)	72.4
Time in Australia ^2,3^		
Recent arrival (≤5 years)	181 (8.5)	18.4
6+ years in Australia	1943 (91.5)	81.6

* Data were from the 2021 census conducted by the Australian Bureau of Statistics [21], ^1^ general population data were from the 2018 national health survey [10], ^2^ the question was not relevant to the 8528 women that were born in Australia, ^3^ general population recent arrivals were from 2016–2021.

**Table 2 healthcare-12-00207-t002:** The association between social determinants and the capacity to access and understand health information.

	Total	Low Health Literacy (≤3)	High Health Literacy (≥4)	*p*
	*n* (%) = 10,652 (100)	*n* (%) = 2546 (23.9)	*n* (%) = 8106 (76.1)	
	M (SD)	M (SD)	M (SD)	
Age (years)	50.55 (14.9)	48.18 (15.7)	53.18 (14.6)	<0.001
	*n* (%)	%	%	
Education				<0.001
No tertiary education	3978 (37.3)	32.1	67.9	
Tertiary education	6674 (62.7)	19.0	81.0	
Self-reported health				<0.001
Good/poor/very poor	6220 (58.4)	28.9	71.1	
Excellent/very good	4432 (41.6)	16.9	83.1	
Financial situation				<0.001
Just getting by/finding it quite difficult/finding it very difficult	2463 (23.1)	38.2	61.8	
Living comfortably/doing alright	8189 (76.9)	19.6	80.4	
First Nations origin				<0.001
First Nations	507 (4.8)	41.4	58.6	
Not First Nations	10,145 (95.2)	23.0	77.0	
Main language				<0.001
Language other than English	435 (4.1)	54.0	46.0	
English	10,217 (95.9)	22.6	77.4	
Country of birth				<0.001
Born overseas	923 (8.7)	39.8	60.2	
Australia	9729 (91.3)	22.4	77.6	
Time in Australia *				<0.001
Recent arrival (≤5 years)	181 (8.5)	50.8	49.2	
6+ years in Australia	1943 (91.5)	26.9	73.1	

* The question was not relevant to the 8528 women who were born in Australia.

**Table 3 healthcare-12-00207-t003:** Multivariable logistic regression analysis of social determinants associated with low health literacy.

	β	S.E. *	df ^1^	*p*	OR ^2^	95% C.I. ^3^
Age (years)	−0.007	0.002	1	<0.001	0.993	0.990–0.996
No tertiary education	0.601	0.049	1	<0.001	1.824	1.658–2.008
Worse self-reported health	0.507	0.052	1	<0.001	1.661	1.501–1.837
Financially disadvantaged	0.578	0.054	1	<0.001	1.783	1.604–1.983
First Nations	0.577	0.101	1	<0.001	1.781	1.460–2.172
Language other than English	1.356	0.104	1	<0.001	3.882	3.167–4.758
Constant	−1.648	0.097	1	<0.001	0.192	

* S.E. = standard error, ^1^ df = degrees of freedom, ^2^ OR = odds ratio, ^3^ C.I. = confidence interval.

## Data Availability

The data presented in this study are available from the corresponding author upon request.

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
