# Peer review of "The Impact of Social Determinants of Health on Australian Women’s Capacity to Access and Understand Health Information: A Secondary Analysis of the 2022 National Women’s Health Survey"

_healthcare, 2024, doi:10.3390/healthcare12020207_

Round 1

Reviewer 1 Report

Comments and Suggestions for Authors

The study sought to investigate associations between social determinants of health and Australian women's ability to access and understand health information. The manuscript was well-organized, sample size is considerable; however, several modifications are required.

The rationale of the study should be explained at the end of the introduction.

The strengths and limitations of the study in the second paragraph of the discussion should be converted to the end of the discussion. 

The discussion part should be shorter, with more references to compare the outcomes with previous studies in other parts of the world.

Additional comments:

The methodology was acceptable. The conclusions were consistent with the evidence and arguments presented, and they addressed the main question.

Comments on the Quality of English Language

Moderate proofreading is required.

Author Response

Dear Reviewer,

We thank you for taking the time to review this manuscript and helpful suggestions to strengthen it. Please find the detailed responses to each comment below and the corresponding revisions to the manuscript. 

Reviewer 2: The rationale of the study should be explained at the end of the introduction.

Authors’ response: We have added additional text elaborating on the rationale behind the study objective.

The final paragraph of the introduction now reads:

Existing literature on the influence of some social determinants of health, on health literacy is limited. The level of health literacy among Aboriginal and Torres Strait Islander peoples, Australia’s First Nations population which represents 3.2% of the Australian population, has yet to be reported, it is assumed to be worse than the health literacy of non-Indigenous peoples [11, 12]. Additionally, health literacy has been shown to differ between men and women [4]. Studies show that women generally have better health literacy or the same level of health literacy as men [9, 10, 13].  Little is known about how the social determinants of health influence the health literacy of women in Australia. Therefore, the aim of this study was to investigate the influence of social determinants of health on Australian women’s capacity to access and understand health information. 

Reviewer 2: The strengths and limitations of the study in the second paragraph of the discussion should be converted to the end of the discussion.

 Authors’ response: We believe that stating the strengths and limitations of the study at the beginning of the discussion provides important context for the interpretation of the findings. Reviewer 3 noted that the discussion section's key aspect was the acknowledgment of limitations, and that we had effectively identified potential drawbacks. 

We would therefore prefer to keep this paragraph where it is but leave it up to the editor to decide the most appropriate place for the strengths and limitations paragraph.

Reviewer 2: The discussion part should be shorter, with more references to compare the outcomes with previous studies in other parts of the world.

Authors’ response: We acknowledge the comment from reviewer 2 concerning the length of the discussion. However, reviewers 3 and 4 endorsed the discussion, appreciating both its length and depth. Consequently, we do not believe that shortening the discussion would improve the paper.

Reviewer 2 Report

Comments and Suggestions for Authors

This manuscript is a well-written description of a cross-sectional study using secondary data that was adapted to evaluate predictors related to social determinants of health and health literacy.  

Introduction:

The introduction is well written and the background is presented in a way that even those that are not primary researchers of SDOH or health literacy can understand the concepts and components of those two important components of public health.  The background is concise and no additional editing would be required.

Methods:

The methods are appropriate for this study however there are two areas within both the dependent and independent variables that could be addressed.  There was no description of how the authors dichotomized the independent variable related to health literacy.  There could be some additional information on why the level was split at <3 or >4?  Other dichotomous variables are straightforward.  The other variable that may need some explanation is the variable related to self-reported health.  This variable could be dichotomized in different ways but the authors should provide a brief recommendation on this rationale.  There is some evidence that how you dichotomize can affect males, it may be less important for females etc. (Bourne, 2009).

Results:

The results section was well described and very straight forward, the tables were useful and were easy to understand.

Discussion:

The discussion section was written appropriately and offered insight into the results without just repeating them.  The key component of the discussion section was the limitations which the authors did a nice job of identifying potential limitations which most readers would identify as they read the manuscript.

Conclusion:

Appropriate, no edits necessary.

Author Response

Dear Reviewer,

We thank you for taking the time to review this manuscript and helpful suggestions to strengthen it. Please find the detailed responses to each comment below and the corresponding revisions to the manuscript. 

Reviewer 3: The methods are appropriate for this study however there are two areas within both the dependent and independent variables that could be addressed.  There was no description of how the authors dichotomized the independent variable related to health literacy.  There could be some additional information on why the level was split at <3 or >4?  Other dichotomous variables are straightforward.  The other variable that may need some explanation is the variable related to self-reported health.  This variable could be dichotomized in different ways but the authors should provide a brief recommendation on this rationale.  There is some evidence that how you dichotomize can affect males, it may be less important for females etc. (Bourne, 2009).

Authors’ response: We have added more details explaining the rationale behind the dichotomization of the independent variable, the health literacy score, and the dependent variable, self-reported health.

            The revised methods section 2.4.2 Dichotomised variables now reads:

Health literacy score

To test for associations between social determinants of health and the capacity to access and understand health information the health literacy score was dichotomised into low (≤3) and high (≥4) health literacy. This split was based on current health literacy literature [6] and the distribution of the data.

Time in Australia

The two categories for time in Australia were based on a report released by the Department of Immigration and Citizenship which classified ‘New Arrivals’ as people who had arrived in Australia within the previous five years [17]. Therefore, the two categories for time in Australia used in this secondary data analysis were ≤5 years and >6 years since arrival in Australia.

County of birth and main language

Country of birth was dichotomised into English-speaking or non-English-speaking countries. This classification was adapted from the ABS [18] and described Australia, the United Kingdom (UK), The Republic of Ireland, the USA, Aotearoa New Zealand, Canada, South Africa, British Overseas Territories, and the US Virgin Islands as the main English-speaking countries. Main language was dichotomised to align with country of birth.

Other dichotomised variables

Education level was dichotomised as tertiary education and no tertiary education. This was based on the Australian Qualification Framework [19]. SRH was dichotomised as excellent/very good and good/poor/very poor, following the method used by the ABS for dichotomising this variable in the 2018 National Health Survey [10] and financial situation was dichotomised as living comfortably/doing alright and just getting by/finding it quite difficult/ finding it very difficult. First Nations origin was also dichotomised as Aboriginal, Torres Strait Islander or Both and not First Nations.

Reviewer 3 Report

Comments and Suggestions for Authors

I would like to thank the authors for their work.

This is an interesting paper, which aims to investigate the influence of social determinants of health on Australian women’s capacity to access and understand health information, through a secondary data analysis of the Jean Hailes National Women’s Health Survey.

The background is robust and up-to-date; however, I would like to suggest that you look further into the rationale behind the study objective.

The methods are consistent with the aim of the study; I would suggest to add, among the statistical analysis section, if the normality of data distribution has been checked, even if the sample is large.

The results are clear, and the discussion is not speculative.

The conclusions are consistent with the results.

Author Response

Dear Reviewer,

We would like to thank you for taking the time to review this manuscript and helpful suggestions to strengthen it. Please find the detailed responses to each comment below and the corresponding revisions to the manuscript. 

Reviewer 4: The background is robust and up-to-date; however, I would like to suggest that you look further into the rationale behind the study objective.

Authors’ response: We have added content providing further explanation for the rationale behind the study.

The final paragraph of the introduction now reads:

Existing literature on the influence of some social determinants of health, on health literacy is limited. The level of health literacy among Aboriginal and Torres Strait Islander peoples, Australia’s First Nations population which represents 3.2% of the Australian population, has yet to be reported, it is assumed to be worse than the health literacy of non-Indigenous peoples [11, 12]. Additionally, health literacy has been shown to differ between men and women [4]. Studies show that women generally have better health literacy or the same level of health literacy as men [9, 10, 13].  Little is known about how the social determinants of health influence the health literacy of women in Australia. Therefore, the aim of this study was to investigate the influence of social determinants of health on Australian women’s capacity to access and understand health information. 

Reviewer 4: The methods are consistent with the aim of the study; I would suggest to add, among the statistical analysis section, if the normality of data distribution has been checked, even if the sample is large.

Authors’ response: Thank you for this suggestion. However, we have already commented on the skewness of the categorical independent variables: level of education, language spoken at home and First Nations status. Please refer to section 3.1. Respondents’ characteristics for the details on the distribution of these variables.

Round 2

Reviewer 1 Report

Comments and Suggestions for Authors

The manuscript has been modified. It has the potential to be published.

Comments on the Quality of English Language

Minor proofreading is required.